# Resistance training prevents right ventricle hypertrophy in rats exposed to secondhand cigarette smoke

Ana Caroline Rippi Moreno[1], Gisele Alborghetti Nai[2], Caroline Pancera Laurindo[1], Karen Cristina Rego Gregorio[1], Tiago Olean-Oliveira[1], Marcos Fernando Souza Teixeira[3], Patricia Monteiro Seraphim[1] *

1 Department of Physiotherapy, School of Sciences and Technology, Sao Paulo State University (UNESP), Presidente Prudente, São Paulo, Brazil, 2 Department of Pathology, University of Western Sao Paulo, (UNOESTE), Presidente Prudente, São Paulo, Brazil, 3 Department of Chemistry and Biochemistry, School of Sciences and Technology, Sao Paulo State University (UNESP), Presidente Prudente, São Paulo, Brazil

* pm.seraphim@unesp.br

**Data Availability Statement:** All relevant data are within the manuscript and its Supporting Information files.

## Abstract

Exposure to secondhand cigarette smoke is associated with the development of diverse diseases. Resistance training has been considered one of the most useful tools for patients with pulmonary disease, improving their quality of life. This study aimed to evaluate the effect of resistance training (RT) on the prevention of thickening of the right ventricle wall of rats exposed to secondhand cigarette smoke. Thirty-two Wistar rats were divided into four groups: Control (C), Smoker (S), Exercised (E) and Exercised Smoker (ES). The smoker groups were exposed to the smoke of four cigarettes for 30 min, twice daily, five days a week, for 16 weeks. The exercised groups climbed on a vertical ladder with progressive load, once a day, five days a week, for 16 weeks. The heart, trachea, lung, liver and gastrocnemius muscle were removed for histopathological analysis. Pulmonary emphysema (S and ES $vs$ C and E, $P < 0.0001$) and pulmonary artery thickness enlargement (S $vs$ C and E, $P = 0.003$, ES $vs$ C, $P = 0.003$) were detected in the smoking groups. There was an increase in the right ventricle thickness in the S group compared with all other groups ($P < 0.0001$). An increase in resident macrophages in the liver was detected in both smoking groups compared with the C group ($P = 0.002$). Additionally, a relevant reduction of the diameter of the muscle fibers was detected only in ES compared with the C, S and E groups ($P = 0.0002$), impairing, at least in part, the muscle mass in exercised smoking rats. Therefore, it was concluded that resistance training prevented the increase of thickness of the right ventricle in rats exposed to secondhand cigarette smoke, but it may be not so beneficial for the skeletal muscle of smoking rats.

## 1. Introduction

Smoking is a preventable disease responsible for a large number of deaths worldwide and approximately 6 million deaths per year [1]. The use of cigarettes exposes users to

**Funding:** The author(s) received no specific funding for this work.

**Competing interests:** The authors have declared that no competing interests exist.

approximately 4720 noxious substances [2], impairing the action and function of several tissues [3, 4]. According to the World Health Organization (WHO), smoking is considered a chronic and epidemic disease and is predicted to cause more than 10 million deaths by 2030 [5].

Secondhand smoke is the combination of two smokes: the mainstream exhaled by smokers plus the burning end of a cigarette (sidestream). Its smoke contains many noxious substances, with hundreds toxic, and some can trigger câncer [6]. Exposure to secondhand cigarette smoke is related to the development of more than 50 diseases, including pulmonary emphysema and chronic bronchitis, which compose chronic obstructive pulmonary disease (COPD) [7]. Smokers are 14 times more likely to develop COPD, and 90% of the deaths caused by COPD occur in smokers or former smokers [8]. COPDs are the fourth leading cause of death in the world, and are expected to reach the third position in 2020 [7]. Socioeconomic costs for the treatment of these and other diseases related to smoking are high, and it is crucial to find tools capable of preventing and/or reducing the ill effects of this habit.

Patients with COPD often have pulmonary hyperinflation [8]. The hyperinflation consists of persistent elevation of pulmonary arterial pressure, which may be caused by increased pressure in the venous and arterial segments of the pulmonary circulation, resulting in increased pulmonary artery thickness [9]. An increase in pulmonary artery thickness is not a specific disease but a pathophysiological condition. However, it can lead to right ventricle overload, hypertrophy and dilatation of the right ventricle free wall developing pulmonary arterial hypertension [10].

Recently, exercise in the form of resistance training (RT) was classified as one of the best and most useful options for the treatment of patients with pulmonary diseases [11, 12]. It is also an effective tool for preventing insulin resistance, controlling arterial and pulmonary hypertension, in addition to improving quality of life [13]. The predominant metabolism in RT is anaerobic, involving adenosine triphosphate and phosphocreatine (ATP-PC) and the glycolytic pathway [14]. This type of metabolism ensures proper muscular endurance associated with the production of maximum strength and power, with pauses for recovery during the performance, among the series [15].

Although several benefits of RT have already been described in the literature, there is no consensus about its real contribution to the prevention of tissue alterations provoked by exposure to secondhand cigarette smoke. The present study aimed to evaluate the effect of resistance training on anatomopathological changes in the right and left ventricles, trachea, lung, gastrocnemius muscle and liver of rats exposed to secondhand smoke.

## 2. Material and methods

### 2.1 Animals

Thirty-two male Wistar rats, aged 45 days, were separated into cages with four animals per cage and kept under a controlled temperature ($23 \pm 2°C$), in a light/dark cycle (12 h/12 h) room [16], with free access to water. The cages were filled with environmental enrichment, and all procedures were carefully done to reduce stress level, and ensure the welfare of the animals [17]. Every day the animals were monitored by observing the food intake behavior, and no alteration was noticed during all procedures with the animals.

Forty grams of commercial chow (Supralab—Alisul, Maringa, PR, Brazil) were offered to each rat, every Monday, Wednesday and Friday. The rest of the chow was weighed and subtracted from the initial offer (40 g) for monitoring. After the adaptation period, resistance training and exposure to secondhand cigarette smoke were performed for 16 weeks.

The weight gain was determined by subtracting the initial weight from the final weight on the euthanasia day.

All procedures complied with the ethical principles of animal research and were approved by the Ethical Committee for Animal Research of the School of Sciences and Technology, Sao Paulo State University, Presidente Prudente (# 02/2017).

## 2.2 Experimental design

The animals were divided into four groups: Control (C / n = 8)—no intervention; Exercised (E / n = 8)—performed RT; Smoker (S / n = 8)—exposed to secondhand cigarette smoke; Exercised Smoker (ES / n = 8)—exposed to secondhand smoke exposure and performed RT.

## 2.3 Smoking exposure

The smoking rats (S and ES) were exposed to secondhand cigarette smoke for 16 weeks [18, 19]. In the first adaptation week, all animals in the S and ES groups were exposed to secondhand cigarette smoke from combustion of 2 cigarettes for 10 min per day. A specific gas detector (ToxiPro® from Biosystems) placed inside the chamber measured 250 ppm (parts per million) of CO (carbon monoxide) during this adaptation period to the secondhand exposure, as described in the literature [20]. After adaptation week, the smoking rats (S and ES) were exposed to further 16 weeks [18, 21]. During the experimental protocol, the smoking rats were exposed to secondhand smoke from the combustion of 4 cigarettes for 30 min long, twice a day, 5 days a week, with 350 ppm of CO / exposure. The referred dose of CO was similar to the previous studies, which avoid the risk of mortality [18–21].

The inhalation system was a custom-built system composed by a closed glass box (100 x 44 x 44 cm), divided in 2 different compartments: one compartment (A) allocated the burning cigarettes, and the second compartment (B) allocated one cage with 4 rats to be exposed to the cigarette smoke. A ventilator was coupled to the lateral wall of the compartment (A) to push the smoke of the smoldered cigarette to the compartment (B). An exhauster was coupled to the lateral wall of the compartment (B) drawing the air through the chamber (Fig 1). Four cigarettes were lit and the complete combustion occurred during 10 minutes with no puff in the compartment (A). Additionally, the rats spent 20 minutes remaining in the compartment (B) inhaling the air saturated with the smoke of the smoldered cigarette in the compartment (A). Although the cigarettes were not pre-balanced with consistent moisture before beginning of the exposure, the sequence of the cages exposed in each day was alternated, to ensure similar characteristics of the air inhaled by all smoking rats until the end of the intervention period. Commercial cigarettes (Malboro Red®, Philip Morris International, Brazil) containing 10mg of tar, 0.8 mg of nicotine, and 10 mg of carbon monoxide during the combustion were used, as already found in the literature [21, 22].

## 2.4 Resistance training

The exercised groups (E and ES) performed the climbing protocol previously reported in the literature by Horberger and Farrar [23], which mimics progressive resistance exercises in humans. Two phases were included: adaptation and experimental.

The adaptation phasis comprised the first 4 days of the intervention, in which the animals adapted to the climbing exercise by being stimulated by a manual stimulus in their rostral portion to climb to a cage present at the top of a ladder, where they were able to rest for 60 seconds. The protocol was repeated 4 x a day.

**2.4.1 Maximum Supported Load (MSL) test.** On the fifth day, the rats performed the maximum supported load test to determine which initial load each animal could lift [24, 25]. A

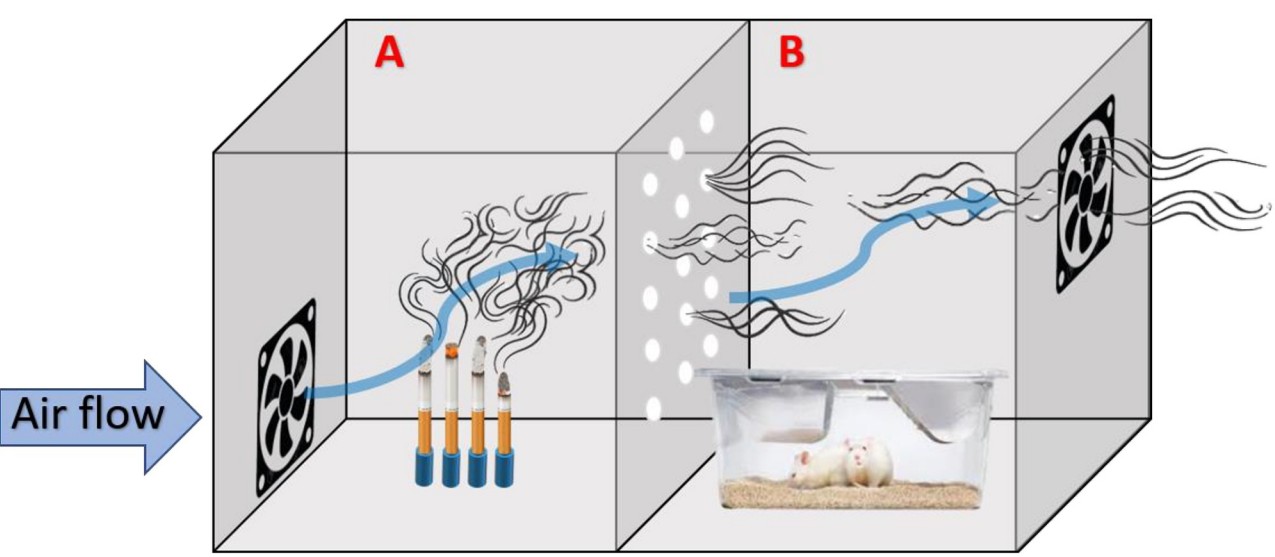

**Fig 1. Custom-built system of chamber enable to expose animals to the cigarette smoke.**

glove containing washers inside was used as load, and it was attached to the tail of the animal by adhesive tape. The initial load was fixed at 75% of body weight for each animal. After each complete climb, 30 grams of weight were added. The test was interrupted if the animal performed more than eight repetitions and if, after three consecutive stimuli, the animal could not perform the climb. The maximum load recorded was the weight corresponding to the last complete climb [23–25] (see S1 Table).

**2.4.2 Experimental phase.** The training consisted of four series of climbs on a ladder, with intervals of 60 seconds, five times a week (on consecutive days), once per day. The training started with 50% of 01 MSL for each animal with the load increasing to 75%, 90% and 100% of 01 MSL every 4 weeks, resulting in an experimental period with 16 weeks of intervention, characteristic of long-term exercise [23–25].

## 2.5 Euthanasia

Anesthesia with intraperitoneal ketamine hydrochloride (60 mg / kg body weight) and xylazine hydrochloride (10 mg / kg body weight) was injected after 12 h of fasting, 24 h after the last RT and smoking sessions for removal of organs (liver, trachea, lung, heart and gastrocnemius skeletal muscle). Euthanasia occurred by exsanguination (Fig 2). Naso-anal length was measured with a ruler and body mass was weighed in the scale for calculation of Lee Index (weight$^{1/3}$/naso-anal length).

## 2.6 Histopathological analysis

Liver, trachea, lung, heart and gastrocnemius skeletal muscle were weighed on a precision scale; the relative weight was calculated [26] (tissue weight/final weight x10$^{-2}$), and then the tissue was stored in 10% buffered formalin. The lungs, heart and trachea were collected together. After withdrawal of the heart, 10% buffered formalin was injected slowly through the trachea for fixation of the pulmonary parenchyma. After 24 h, fragments of the right and left lung of each animal were removed. The heart was coronally sectioned at the level of the atria and ventricles, and the liver, trachea, and gastrocnemius muscle were sectioned in transverse sections.

**Fig 2. Timeline of the interventions performed over 18 weeks.** 1 MSL = maximum supported load; RT = resistance training; min = minutes.

The sections were submitted to normal histological processing, with inclusion in paraffin (Dynamics Analytical Reagents, São Paulo, Brazil). Three serial cuts of 5 μm with a spacing of 15 μm were performed on all slides. Some slides were stained by hematoxylin-eosin (HE) (Dolles, São Paulo, Brazil), others by Masson's (Merck KgaA, Darmstadt Germany) and Alcian Blue—PAS Staining (Merck, Germany), as described in detail below.

The histopathological analysis was blinded and performed by a single experienced observer using an optical microscope (NIKON Labophot, Japan). The original stained image was captured by a Leica photomicroscope (Leica Microssistems, Switzerland) using ImageJ® software from the National Institute of Health (NIH, USA). The parameters evaluated were as follows:

1. Two stainings were performed for the trachea: the first was HE staining to identify inflammatory infiltrate (0 = absent, 1 = mild, 2 = moderate, 3 = severe) and the inflammatory cell type (polymorphonuclear and / or mononuclear). The second was Alcian Blue—PAS staining, with ten high-power fields (HPFs) and a magnification of 40x; 10 photos corresponding to approximately 1 mm$^2$ were taken to identify the number of goblet cells [27, 28].

2. Two stainings were performed for the lung: the first was HE staining to identify interstitial inflammatory infiltrate (0 = absent, 1 = mild, 2 = moderate, 3 = severe); the inflammatory cell type (polymorphonuclear and / or mononuclear) and location (intra-alveolar, interstitial and peribronchial); tissue congestion (0 = absent, 1 = mild, 2 = moderate, 3 = severe); interstitial fibrosis (0 = absent, 1 = focal, 2 = diffuse); and emphysema [0 = absent, 1 = focal (compromising only part of the pulmonary parenchyma), 2 = diffuse (compromising all lung parenchyma)]. The second was Alcian Blue-PAS staining for analysis of pulmonary artery thickness, with two photos per animal and two measurements per photo (magnification of 200x) [28, 29]. Pulmonary emphysema was diagnosed when the alveolar spaces were enlarged and the alveolar septa retracted.

3. Two stainings were used in the heart: the first was HE staining to measure the thickness of the left and right ventricular free walls and the interventricular septum [28, 29]. One photo of each area (magnification of 100x) was taken with two measurements per picture. The second staining was by Masson's trichrome staining for quantification of collagen fibers, as described in detail below [30].

4. HE staining was used in the gastrocnemius skeletal muscle. Sections were taken from the central area of the muscle. The analysis of the diameter of the muscle fibers was performed in 50 intact fibers in the HPF photos [31].

5. HE staining was used in the liver to identify tissue congestion (0 = absent, 1 = mild, 2 = moderate, 3 = severe); inflammatory interstitial infiltrate (0 = absent, 1 = mild, 2 = moderate, 3 = severe) and the type of inflammatory cell present (polymorphonuclear and / or mononuclear); necrosis (0 = absent; 1 = present); cholestasis (0 = absent, 1 = present); presence and type of steatosis (0 = absent, 1 = present: microvesicular and / or macrovesicular). The Kupffer cells (resident macrophages) were counted in 10 HPF, corresponding to approximately 1 mm$^2$ in each sample.

## 2.7 Collagen density (Fractal dimension)

To perform the analysis of the fractal dimension of the right and left ventricles, the slides stained with Masson's trichrome were photographed (one photo for each area for each animal) and passed through the binarization process to read and analyze the fractal dimension with the box-counting method, using free ImageJ (NIH) software (http://rsbweb.nih.gov/ij/).

ImageJ software performs box counting in two dimensions, allowing the quantification of the distribution of pixels in the space. The fractal analysis of the histological slides is the relation between the resolution and the evaluated scale: DF = (Log Nr / log r-1), with Nr being the amount of equal elements needed to fill the original object and r being the scale applied to the object. Therefore, the fractal dimension calculated with ImageJ software will always be between 0 and 2, relative to the density of cardiac collagen [30].

## 2.8 Statistics

The results are expressed as the mean ± standard error of the mean (SEM). Two-way ANOVA was used for the quantitative results, with Tukey's as a posttest, and Kruskal-Wallis was used for analysis of categorical variables. *P* values less than 5% were considered statistically significant. GraphPad Prism software version 6.0 was used.

## 3. Results

The smoking groups had lower body mass gains compared to the C group (*P* = 0.005) but a similar Lee Index. However, only the ES group showed a significant reduction in feed intake compared to the C and E groups (*P* = 0.0005), as well as a reduction in final body weight compared to the C group (*P* = 0.001) (Table 1).

The trachea showed no inflammatory infiltrate or change in goblet cell counts (see S1 Fig). However, although the pulmonary tissue showed no inflammation, tissue congestion or

**Table 1. Characteristics of the animals.**

|  | C | E | S | ES |
|---|---|---|---|---|
| **Body mass gain (g)** | 230.50±10.80 | 217.25±13.22 | 184.75±6.35* | 169.12±9.69# |
| **Final body weight (g)** | 482.62±12.71 | 463.87±15.30 | 435.12±6.80 | 414.01±16.04* |
| **Feed consumption (g)** | 29.52±0.58 | 29.33±0.60 | 28.48±0.42 | 26.59±0.40# |
| **Lee Index** | 320.19±3.34 | 317.49±3.13 | 317.48±2.74 | 312.63±2.04 |
| **Skeletal muscle gastrocnemius (g)** | 2.8±0.1 | 2.7±0.1 | 2.7±0.1 | 2.6±0.1 |

Lee Index (weight $^{1/3}$/naso-anal length in cm). Body mass gain (final body weight—initial body weight): *P = 0.005 vs C; #P = 0.005 vs C and E. Final body weight: *P = 0.001 vs C. Feed consumption (day / animal): #P = 0.0005 vs C and E. Values expressed as the mean ± SEM (n = 8 / group).

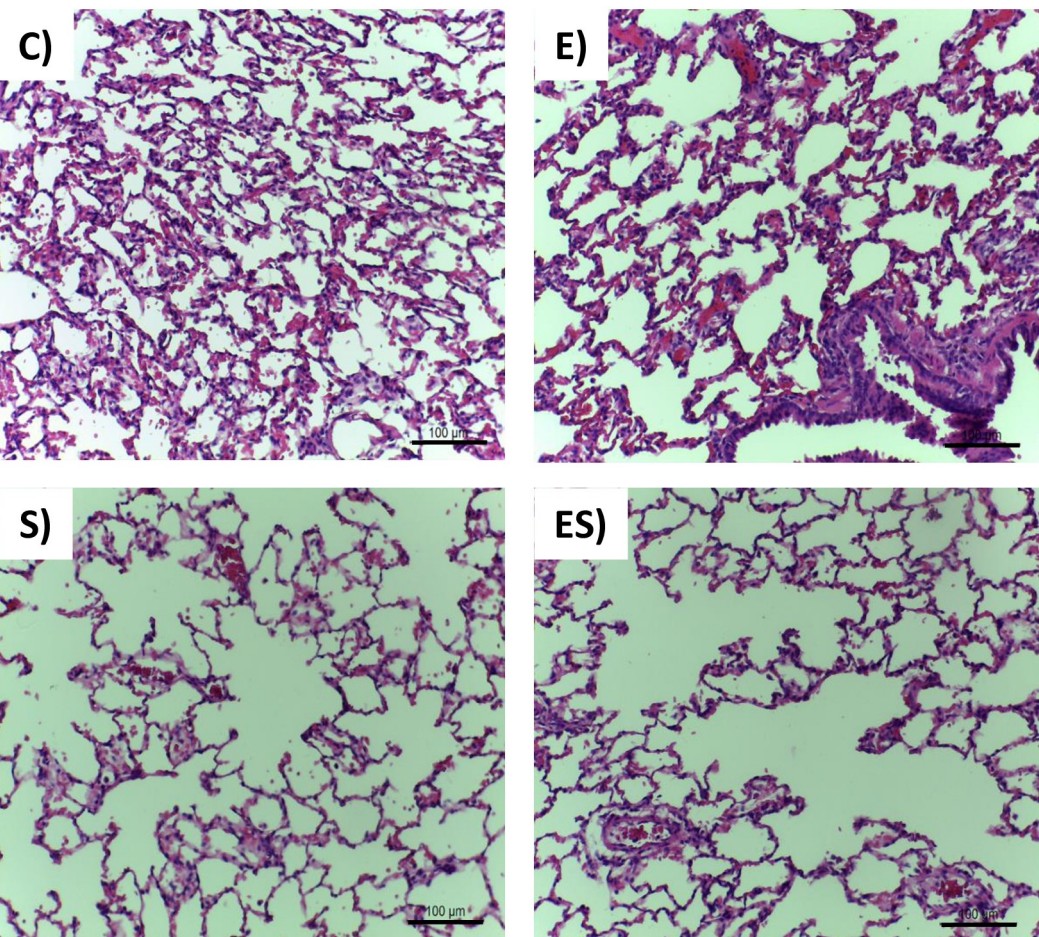

**Fig 3. Histopathological assessment of the lung with HE staining.** Upper left image—Typical Pulmonary parenchyma without alteration in C group. Upper right image—Typical Pulmonary parenchyma without alteration in E group. Bottom left image—Typical Pulmonary emphysema in S group. Note the dilated alveolar spaces. Bottom right image—Typical Pulmonary emphysema in ES group. Note the dilated alveolar spaces.

interstitial fibrosis, pulmonary emphysema was observed in the smoking groups (Fig 3) (S and ES, $P < 0.0001$). Almost all smoking animals, sedentary and exercised, presented diffuse emphysema, except one ES rat.

In the evaluation of pulmonary artery thickness, there was a significant increment in the smoking groups, with an increase in the S group compared to the control groups (C and E) and in the ES group compared to the C group ($P = 0.003$) (Fig 4).

The S group presented increased right ventricle thickness compared to all other groups (Fig 5) ($P < 0.0001$). However, no significant alteration was observed in the evaluation of left ventricle thickness or septum. In the fractal analysis of both ventricles, no significant alteration was detected among the groups (see S2 Fig).

The diameter of the fibers of the gastrocnemius muscle was significantly reduced in the ES group compared to all other groups ($P = 0.0002$) (Fig 6).

In the liver, no inflammatory infiltrate, necrosis, cholestasis, or steatosis were observed in the groups (Fig 7). However, there was a significant increase of approximately 30% in the

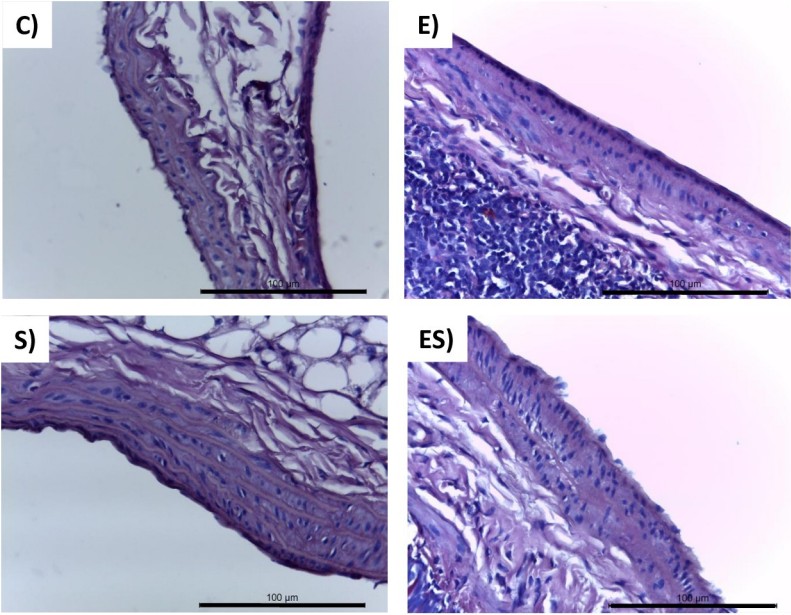

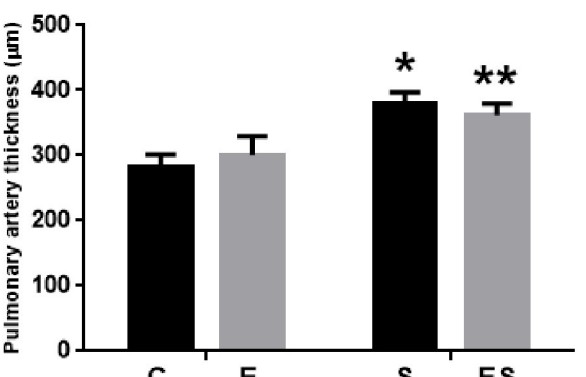

**Fig 4. Typical histopathological evaluation of the pulmonary artery thickness with Alcian blue—PAS.** Upper left image—C group, Upper right image—E group, Bottom left image—S group and Bottom right image—ES group. Graph: Pulmonary artery thickness. $^*P = 0.003$ vs C and E groups, $^{**}P = 0.003$ vs C. Values are expressed as the mean ± SEM (n = 8 / group).

smoking groups (S and ES, $P = 0.002$) compared to the C group in the Kupffer cell counting, which indicates the number of resident macrophages of this tissue.

## 4 Discussion

In the present study, we observed that exposure to secondhand cigarette smoke decreased body mass, provoked pulmonary emphysema, and increased pulmonary artery thickness and the amount of resident macrophages in the liver. Resistance training in the smoking rats reduced the final body weight, which was associated with lower feed intake, reduced muscle

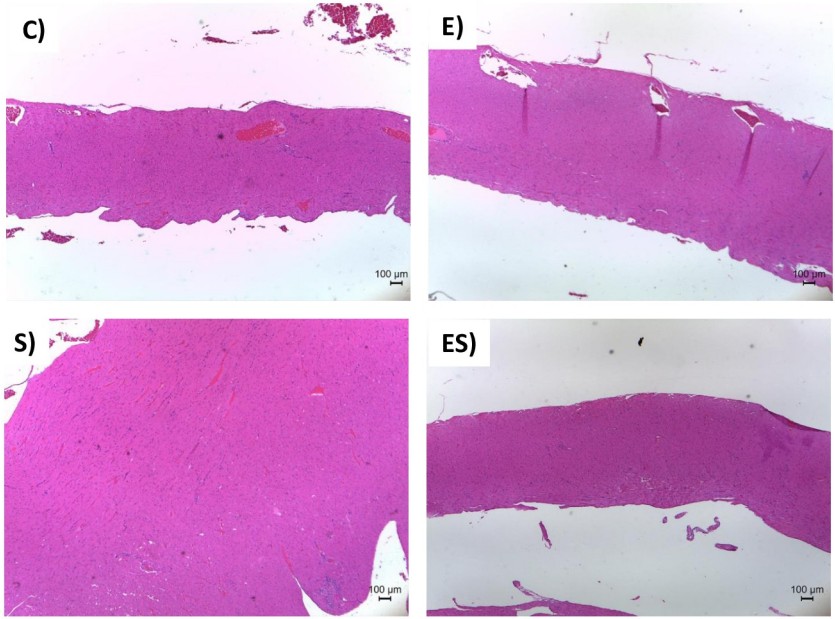

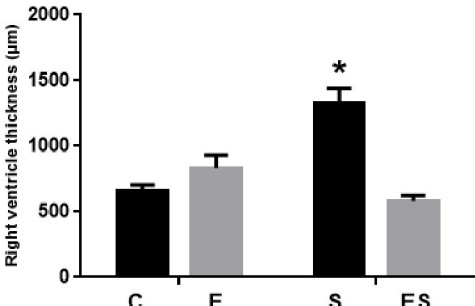

**Fig 5. Histopathological evaluation of the right ventricle thickness with HE staining.** Upper left image—C group, Upper right image—E group, Bottom left image—S group and Bottom right image—ES group. Graph: Thickness of the right ventricle (in μm). * *P* <0.0001 vs C, E and ES. Values expressed as the mean ± SEM (n = 8 / group).

fiber diameter of the gastrocnemius muscle, and maintenance of right ventricle thickness, which likely prevented pulmonary arterial hypertension development.

Cigarettes are the main licit drug consumed worldwide. It is estimated that there are 1,2 billion smokers worldwide and that one in five people has this harmful habit [5]. Smoking, in addition to causing dependence, is related to the appearance of several chronic diseases in the lung, heart, liver and skeletal muscle, in addition to malnutrition and emotional illness [3, 4].

A systematic review examined the effects of cigarette smoke on the body weight of smokers, concluding that low- and medium-intensity smokers (up to 1 pack per day) have a significant reduction of body weight due to nicotine causing an increase in metabolic rate and a reduction in caloric absorption and appetite [32]. In the present study, rats exposed to secondhand smoke had reduced body mass gain. In the ES group, a significant reduction in weight gain

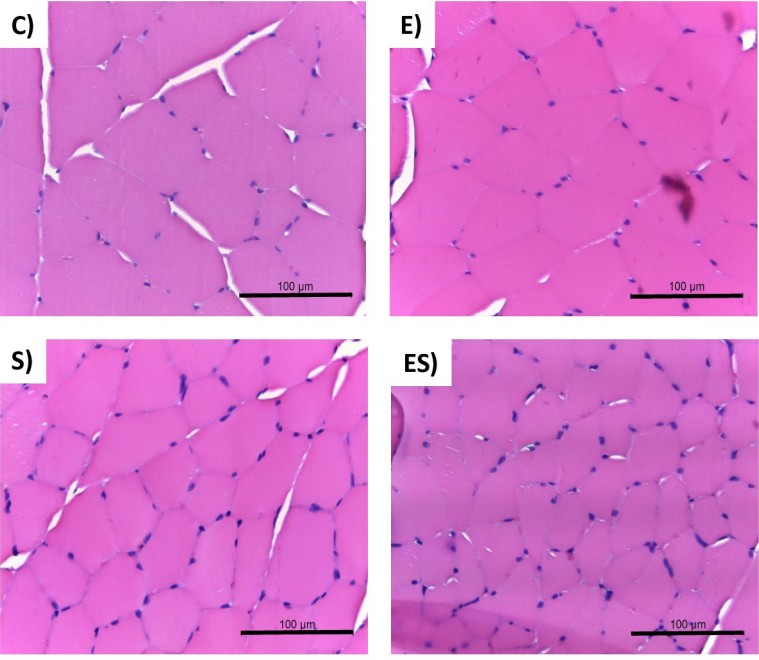

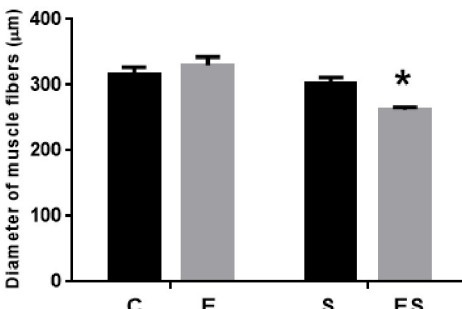

**Fig 6. Typical histopathological evaluation of the gastrocnemius muscle with HE staining.** Upper left image—C group, Upper right image—E group, Bottom left image—S group and Bottom right image ES group. Graph: Diameter of gastrocnemius skeletal muscle fibers (in μm). * $P = 0.0002$ vs C, E, S. Data are mean ± SEM, values expressed as μm / animal (n = 8 / group).

was observed compared to both control and exercised groups, accompanied by a significant reduction in final body weight and feed consumption, as described in the literature [32].

Among the COPD spectrum is pulmonary emphysema, which is characterized by the destruction of the alveolar walls and an increase in the air spaces distal to the terminal bronchioles [33]. Patients with COPD often present nutritional changes due to smoking, resulting in a weight reduction, as well as negative prognoses: survival is reduced by 13 years on average [34]. In the present study, the presence of pulmonary emphysema was detected in both groups of smokers (S and ES) groups that had reduced weight gain. In addition, the pulmonary emphysema detected was in agreement with the study by Kozma et al. [7], who concluded that exposure to secondary cigarette smoke is capable of developing a model of oxidative lung

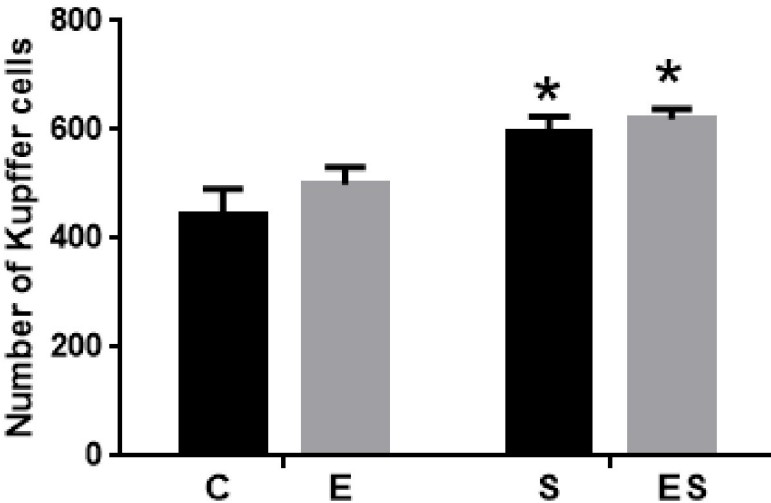

**Fig 7. Number of Kupffer cells in the liver (per mm$^2$).** * $P$ = 0.0003 vs C group. Values of the sum of 10 fields expressed as the mean ± SEM (n = 8 / group).

injury and inflammation, accelerating functional and morphological alterations, and limiting gas exchange [35].

It is known the worsening of pulmonary emphysema can impairs the functional capacity of skeletal muscle, such as reduction of type I motor units; atrophy of motor units; reduced capillarity and altered levels of metabolic enzymes [7]. Thus, COPD is can considered a systemic disease that is capable of resulting in musculoskeletal changes, such as weakness, muscular dysfunction and weight reduction, due to chronic oxidative stress, which also contributes to reduced caloric intake [36]. The gastrocnemius muscle was evaluated in the present study because it has mixed skeletal muscle fibers, with characteristics of type I and II motor units. This muscle is considered the prime driver of extension at the hip and knee joints and of plantarflexion at the ankle, all of which are essential movements for stair climbing [37]. A reduction in the diameter of the gastrocnemius muscle fiber was detected in the ES group. Additionally, ES rats showed reduced final body weight, reduced feed consumption, lower weight gain, pulmonary emphysema and increased macrophages in the liver. We can suggest that the reduction in the diameter of the muscle fiber can be associated with the diagnosis of pulmonary emphysema, which as previously mentioned [36], can develop negative skeletal muscle changes. In this case, although we have not studied any type of reactive oxygen species (ROS), we know that the practice of physical activity and smoking habit can develop a state of chronic oxidative stress [36], which would impair the development of muscle fibers. Thus, we suggest that the intensity of the training of smokers should be evaluated, and that RT should be performed with considerable caution in smokers.

In addition to peripheral muscle changes, exposure to smoking can also cause changes in respiratory muscles, which are responsible for pulmonary hyperinflation, a characteristic of COPD patients [38]. Pulmonary hyperinflation can change the shape of the chest wall and reduces the number of muscle fibers in the diaphragm, causing the muscle to work under increased mechanical load due to airflow limitation [39]. Thus, changes in the structure of the diaphragm in smokers include an increase in type I motor units and a decrease in type II motor units, possibly increasing the oxidative capacity of the muscle fibers of this muscle [40]. This is an aerobic adaptation of the diaphragm, which is insufficient to restore a normal contraction force, which may result in a physiopathological condition of hyperinflation [40].

Pulmonary hypertension (PH) is a hemodynamic disorder defined by an abnormal increase in pulmonary arterial pressure (PAPm), classified as PAPm $\geq$ 25 mmHg [41]. The three main features of PH are as follows: 1—dysfunctional pulmonary hemodynamics (i.e., exacerbated vasoconstriction and reduced vasodilation); 2—structural changes in the pulmonary vascularization (i.e., wall remodeling and hypertrophy), leading to elevated pulmonary arterial pressure; and 3- long-term right ventricular (RV) pressure overload and subsequent RV failure [42]. In the present study, we observed a significant increase in the pulmonary artery thickness in both groups of smokers. The smoking but not exercised rats developed an increase in the thickness of the pulmonary artery and of the right ventricle. On the other hand, although RT had not prevented the increase in the thickness of the pulmonary artery, it seemed to prevent the thickening of the right ventricle. It is likely that although RT in smoking rats had not prevented the second stage of PH (remodeling of the pulmonary artery wall), trained smoking rats have not shown an increase in RV, causing a delay in the prognosis of PH. This suggests that the main causal factor in this model is the lack of resistance training, which can contribute for the triggering of pulmonary hypertension in the future. We did not perform right cardiac catheterization (RCC) and transthoracic echocardiography (ET), which are relevant tools for the detection and screening of PH [41]. Further studies can be developed with the intention of verifying whether resistance training prevents the development of PH using CCD and ET techniques.

Increasingly in the literature, it is shown that liver and heart are two organs that can influence each other. There are heart diseases striking the liver, liver diseases striking the heart, and diseases striking the heart and the liver simultaneously [43]. In the present study, when we analyzed the increase of hepatic resident macrophages in the smoking groups (S and ES), we also noticed that although this group did not present steatosis and/or inflammatory infiltrate in the liver, there was innate immune system action, which is always present, ready to defend against microorganisms and eliminate damaged cells [44]. In this case, hepatocytes can have been injured by exposure to secondhand smoke, once smoking can affect organs even that it has no direct contact with cigarette smoke in fact, as liver. Smoking can provoke three effects on the liver: oncogenic effects, indirect or direct toxic effects and immunological effects [45] by agents with cytotoxic potential released during the combustion. These cytotoxic agents trigger oxidative stress, lipid peroxidation activating stellate cells and development of fibrosis in the liver [46]. Although we did not find fibrosis in our model, we find significant increased Kupffer cells after cigarette smoke exposure, suggesting anormal condition in the liver of smoking rats. Once heart was impaired in sedentary smoking rats only, it is likely that this study has been performed in the beginning of the emergence of the complications of the hepatocardiac disease [47]. Further studies characterizing the type of macrophages in the liver and analyzing the ROS reactive oxygen species in smokers could contribute to a better understanding of the mechanisms involved in resistance training effects in this model.

In summary, we suggest that despite the benefits of RT, caution is required considering the intensity of protocol of strength training in smokers, since RT reduced the muscle fibers in the gastrocnemius muscle, which did not occur with nonsmokers. However, we pointed out although resistance training did not prevent the development of pulmonary emphysema, an increase of macrophages in the liver and pulmonary artery hypertrophy, as observed in secondhand smoking rats, helped to prevent increased right ventricle thickness.

## Supporting information

**S1 Fig. Number of goblet cells in the trachea (per mm2).** Values of the sum of 10 fields expressed as the mean ± SEM (n = 8 / group).
(PPTX)

**S2 Fig. Histopathological evaluation of the heart. In A, thickness of the left ventricle (in µm). In B, thickness of the septum (in µm).** In C, fractal dimension of the left ventricle (Log Nr / log r-1). In D, fractal dimension of the right ventricle (Log Nr / log r-1). No significant alteration was observed. Values expressed as the mean ± SEM (n = 8 / group). (PPTX)

**S1 Table. MSL weight—Details exercise capacity and MSL weight for E and ES groups in the beginning of the resistance training protocol.** (DOCX)

**S1 File.** (DOCX)

## Author Contributions

**Conceptualization:** Ana Caroline Rippi Moreno, Gisele Alborghetti Nai, Marcos Fernando Souza Teixeira, Patricia Monteiro Seraphim.

**Formal analysis:** Ana Caroline Rippi Moreno.

**Investigation:** Ana Caroline Rippi Moreno.

**Methodology:** Ana Caroline Rippi Moreno, Tiago Olean-Oliveira, Patricia Monteiro Seraphim.

**Project administration:** Gisele Alborghetti Nai, Patricia Monteiro Seraphim.

**Resources:** Patricia Monteiro Seraphim.

**Supervision:** Patricia Monteiro Seraphim.

**Writing – original draft:** Ana Caroline Rippi Moreno, Gisele Alborghetti Nai, Caroline Pancera Laurindo, Karen Cristina Rego Gregorio.

**Writing – review & editing:** Marcos Fernando Souza Teixeira, Patricia Monteiro Seraphim.

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
