## [Decision Letter · Decision Letter 0]

6 Feb 2020

PONE-D-19-31709

Resistance training prevents right ventricle hypertrophy in rats exposed to secondhand cigarette smoke

PLOS ONE

Dear Dr Seraphim,

Thank you for submitting your manuscript to PLOS ONE. After careful consideration, we feel that it has merit but does not fully meet PLOS ONE’s publication criteria as it currently stands. Therefore, we invite you to submit a revised version of the manuscript that addresses the points raised during the review process.

I agree with Reviewer 1 that you should name the brand of cigarettes.  There would only be a conflict of interests if you had some financial or other relationship with the manufacturer, that would need to be disclosed.

We would appreciate receiving your revised manuscript by Mar 21 2020 11:59PM. To enhance the reproducibility of your results, we recommend that if applicable you deposit your laboratory protocols in protocols.io, where a protocol can be assigned its own identifier (DOI) such that it can be cited independently in the future. For instructions see: http://journals.plos.org/plosone/s/submission-guidelines#loc-laboratory-protocols

We look forward to receiving your revised manuscript.

Kind regards,

Stanton A. Glantz

Academic Editor

PLOS ONE

Journal Requirements:

2. As part of your revision, please complete and submit a copy of the ARRIVE Guidelines checklist, a document that aims to improve experimental reporting and reproducibility of animal studies for purposes of post-publication data analysis and reproducibility: https://www.nc3rs.org.uk/arrive-guidelines. Please include your completed checklist as a Supporting Information file. Note that if your paper is accepted for publication, this checklist will be published as part of your article.

Reviewers' comments:

Reviewer's Responses to Questions

**Comments to the Author**

1. Is the manuscript technically sound, and do the data support the conclusions?

Reviewer #1: Partly

Reviewer #2: Partly

2. Has the statistical analysis been performed appropriately and rigorously? 

Reviewer #1: Yes

Reviewer #2: Yes

3. Have the authors made all data underlying the findings in their manuscript fully available?

Reviewer #1: Yes

Reviewer #2: No

4. Is the manuscript presented in an intelligible fashion and written in standard English?

Reviewer #1: Yes

Reviewer #2: No

5. Review Comments to the Author

Reviewer #1: This manuscript by Moreno et al. reports that RV hypertrophy resulting from secondhand smoke exposure of rats can be prevented by resistance training (RT). The finding that RT completely prevented the otherwise substantial increase in wall thickness is interesting and supported by the data, and suggests that not only smokers, but people exposed frequently to high levels of secondhand smoke, would gain a health benefit from an RT exercise program in addition to the expected general health benefits of exercise. It’s less clear how to interpret the change in skeletal muscle fiber diameter. The histology is convincing and the main conclusions of the paper are solid. I do have many concerns, however, including lack of sufficient information about the smoke exposure, as follows.

Major concerns:

The authors state in their response to previous evaluation that they have opted to not identify the commercial cigarette brand that they have used, to avoid conflict of interest. This is not acceptable; they need to identify the cigarette because not all commercial cigarettes are the same; it’s important information. There should be no concern about conflict of interest here; identification of cigarettes used in such studies is routinely done and the only potential conflict of interest that I can see is if one of the authors works for or is funded by the tobacco company in question or the tobacco industry in general; is this is the case, then it needs to be disclosed but the cigarette still needs to be identified.

There are some essential details about the smoke exposure that are missing and should be added. Please provide some measure of smoke concentrations in the exposure chamber, as either total particulate matter or respirable suspended particles (<2.5 µm) (and potentially CO if that was measured). Is the smoking system a commercial system (if so, which one) or a custom built system? I don’t entirely follow the part about the chamber used for placement of a lit cigarette; despite describing airflow to bring smoke to the rats, there’s no mention of smoke being sucked through the cigarette to generate the smoke (what puff protocol was used; e.g., puff volume, puff duration, puff frequency). If no smoke was sucked through the cigarettes, and they were simple smoldering in place such as sitting in an ashtray, then that is a different model from what is typically used for secondhand smoke and should be explained. Are the multiple cigarettes all lit simultaneously or successively? (I’m guessing it was successively one at a time, since a cigarette probably can’t last for a whole 30 minutes, but please clarify.) Were they pre-equilibrated to consistent humidity levels before use, as is common practice?

I understand that table 1 is being provided to address concerns about the weight bearing protocol, but it looks more like Results than Methods to me.

Discussion: Regarding the sentence starting on line 324: “It is known that exposure to secondhand smoke intensifies the harmful effects of the smoke due to the absence of a filter; secondhand smoke has nicotine and CO2 concentrations three times higher than smoke experienced 336 by a smoker, and it has 50 times more carcinogenic substances (6,7).” I am not aware that this is the case, although it is popularly assumed. To the best of my knowledge, the filter does not filter out very much of the material that gets deep into the airways, and the mainstream smoke inhaled by a smoker is far more concentrated than secondhand smoke. (By CO2, did they mean CO?) They cite reference 6 and 7; 6 is a meta-analysis and 7 is a study about cluster headaches and neither supports the statement. I would remove this highly questionable sentence. This in no way minimizes the importance of secondhand smoke as a major source of adverse health effects.

Minor comments

While the nomenclature of SC, EC, SS, and ES makes sense, it’s REALLY hard to follow because S stands for both Sedentary and Smoker. Yes, it matters if it’s the first or second letter of the pair, but it just makes it much less intuitive to follow along for the reader. Can the authors come up with a similarly informative but more reader-friendly way of denoting the permutations?

Abstract line 37: it will be much easier to follow if a comma is added after EC. Otherwise, it’s not immediately clear whether EC belongs to the first comparison or the second. Even then, it’s a confusing line to read.

The sentence starting on abstract line 41 would be more useful if there was an indication of what the finding means (i.e., tell the reader if reduction of the muscle fiber diameter good or bad).

I rarely comment on keywords, but it seems to me that including “tobacco” as a keyword would be useful.

Line 57: It’s not accurate to say that the “consumption” of a cigarette is divided into particulate and gas phases; consumption is the verb. Better to say cigarette smoke consists of particulate and gas phases. Actually, it may be a language issue, but the entire paragraph is problematic, and references 6 and 7 seem like odd choices to use here. Are the authors attempting to describe the mainstream smoke that is inhaled through the filter and the sidestream smoke that is generated (not “exhaled”) by the burning tip? This is all irrelevant to gas or particle phases; the phrase on line 58 that the gas phase consists of two types of smoke does not make sense. I suggest that they replace this paragraph (and get rid of those two references) with a simple statement that the smoke inhaled by smokers is mainstream smoke, the smoke generated by the burning tip is sidestream smoke, the secondhand smoke in the air around smokers consists of a mixture of mostly sidestream smoke with some exhaled mainstream smoke, and that secondhand smoke contains high concentrations of noxious components.

Line 71: In “…it is fundamental to find tools…”, fundamental should probably be replaced by something like crucial or important.

Line 74: “pulmonary pressure”: do they mean pulmonary arterial pressure or increased pressure in the lungs?

Line 80: I suggest clarifying the first time they say “resistance training” to make it more obvious to readers who aren’t familiar with the topic, especially given that “insulin resistance” shows up a few words later. Perhaps “Recently, exercise in the form of resistance training (RT) was classified…”

In Table 1 in the Methods; please add units (presumably g).

Line 237, I think “tricomium” is supposed to be “trichrome”

Line 257: what is a “Lee index”? Please clarify in the text. It’s in the figure legend but the reader has not yet encountered that when the term is first used.

Line 325: Cigarette smoking is not a drug; cigarettes are the drug.

Line 414: “strength” is misspelled.

In Figure 1, the word “cigarettes” is misspelled twice, and it is missing a space in “1x day(4 days)”

The legends for Figures 2-6 include magnification denotations of the form “100x” etc., which are meaningless unless the final print size of the figure is known and not useful when viewing on a monitor. Scale bars are better, and the figures actually have scale bars already but they are much too small to be seen, so it will be a simple matter to make standard visible scale bars based on them. If a magnification number is desired, please state as “10x objective” etc.

In figures 2-5, where quartets of histology panels are shown, it will be much simpler for the reader if panels are labeled as ES, SS, etc. rather than a, b, etc.; it saves the reader from having to look at the legend to understand the figure.

I suggest making the asterisks in the bar graphs larger; they are tiny and look like lint on the computer monitor.

Reviewer #2: The manuscript by Seraphim et al. aims at determining the effect of resistance training on various parameters in rats exposed to second hand cigarette smoke. Their main conclusion is that the training prevents right ventricle hypertrophy. They provided data that are somehow related to development of right ventricle hypertrophy such as pulmonary artery thickness and lungs histopathology, as well as some supportive information such as body mass gain, food consumption, and masses of the different tissues that they included in their study. Not being an expert on the field, this reviewer cannot say how novel and interesting the findings are for the scientific community. However, there are definitely a number of issues that have to be addressed before the manuscript can be considered for publication.

General observations:

1. This reviewer was invited as an animal experiment ethics specialist. In this respect, the study is fine; there seems to be no ethical concern.

2. Although a language correction has been performed on the manuscript, clarity of the text is still below publishable quality. To illustrate it, this reviewer performed editing of a 1.5 pages section of Materials and Methods. a/ line 120-122: “The smoking rats (SS and ES) were exposed to secondhand cigarette smoke (19) for 16 weeks, which was predetermined to characterize long-term exposure (20).” The second part of the sentence may mean that 16 weeks of exposure is considered as a long term one. b/ line 125: Change “characterizing” to “representing”. c/ line 138: Cigarettes never contain carbon monoxide, which is a gas. The indicated 10 mg should be the average dose a smoker receive during a smoke. d/ line 153-154: “This test was performed only in this moment.” ???? Better delete.

Specific comments:

1. The overall design of the study indicates that more results have been anticipated than have actually been resulted. That creates some problems. The organ masses were properly presented as not significantly altered in line 260. However, according to PLOS ONE’s publishing policies “data not shown” is not permissible. Either the data should be presented, or the observation should be omitted. In this case, I suggest presenting the data as a supplementary table.

2. line 271: On what basis was pulmonary emphysema diagnosed? Put it in the Methods section.

3. Fig 4: Exercise increased right ventricle thickness in non-smoker rats while decreased it to BELOW THE CONTROL LEVEL in smoker ones. These data are very hard to understand.

4. The number of Kupffer cells in the liver is presented in Fig 6. Either move this figure to the supplementary section or explain in the Discussion section the relevance of the data presented in it to right ventricle hypertrophy.

5. The Discussion section is poor. Please, interpret the data presented in the Results section regarding right ventricle hypertrophy and related studies on the field. Also, please avoid drawing conclusions on unsupported data.

6. PLOS authors have the option to publish the peer review history of their article (what does this mean?). If published, this will include your full peer review and any attached files.

Reviewer #1: No

Reviewer #2: No

---

## [Author Response · Author response to Decision Letter 0]

2 Apr 2020

Dear Stanton A. Glantz, 

Firstly, we would like to thank for the appreciattion and comments in oour paper titled “Resistance training prevents right ventricle hypertrophy in rats exposed to secondhand cigarette smoke”. The review was done, and the authors accepted the suggestions as well as implemented the information needed.

The answers for the reviwers´s questions are above, as following:

Reviewer #1:

1- “The authors state in their response to previous evaluation that they have opted to not identify the commercial cigarette brand that they have used, to avoid conflict of interest. This is not acceptable; they need to identify the cigarette because not all commercial cigarettes are the same; it’s important information. There should be no concern about conflict of interest here; identification of cigarettes used in such studies is routinely done and the only potential conflict of interest that I can see is if one of the authors works for or is funded by the tobacco company in question or the tobacco industry in general; is this is the case, then it needs to be disclosed but the cigarette still needs to be identified.”

Answer: The commercial cigarette brand was added to the text in Material and Methods Section of the manuscript.

2- “There are some essential details about the smoke exposure that are missing and should be added. Please provide some measure of smoke concentrations in the exposure chamber, as either total particulate matter or respirable suspended particles (<2.5 µm) (and potentially CO if that was measured). Is the smoking system a commercial system (if so, which one) or a custom built system? I don’t entirely follow the part about the chamber used for placement of a lit cigarette; despite describing airflow to bring smoke to the rats, there’s no mention of smoke being sucked through the cigarette to generate the smoke (what puff protocol was used; e.g., puff volume, puff duration, puff frequency). If no smoke was sucked through the cigarettes, and they were simple smoldering in place such as sitting in an ashtray, then that is a different model from what is typically used for secondhand smoke and should be explained. Are the multiple cigarettes all lit simultaneously or successively? (I’m guessing it was successively one at a time, since a cigarette probably can’t last for a whole 30 minutes, but please clarify) Were they pre-equilibrated to consistent humidity levels before use, as is common practice?”

Answer: The inhalation system was a custom built system composed by a closed glass box (100 x 44 x 44 cm), divided in 2 different compartments: one compartiment for allocation of the cage with 04 rats to be exposed to the cigarette smoke, and the other for the allocation of the cigarettes to be burned. A compressor of 10 L/min-air was coupled to the cigarette compartment to push the smoke to the other compartment of the box. The compartment where the animal was allocated presented a hole for the exhaust of the smoke. Four cigarettes were lit and the complete combustion occurred during 10 minutes, however the rats spent 30 minutes inside this system. So, during the 10 first minutes inside the chamber, the cigarettes were burned, but in the rest of the time the rats were exposed only to the polluted air of the environment. The dose used in the smoke exposure is equivalent to 10 to 20 cigarettes for a chronic human smoker.

4) “I understand that table 1 is being provided to address concerns about the weight bearing protocol, but it looks more like Results than Methods to me.”

Answer: Table 1 was included as Supplementary material – see Suppl 1.

Discussion: Regarding the sentence starting on line 324: “It is known that exposure to secondhand smoke intensifies the harmful effects of the smoke due to the absence of a filter; secondhand smoke has nicotine and CO2 concentrations three times higher than smoke experienced 336 by a smoker, and it has 50 times more carcinogenic substances (6,7).” I am not aware that this is the case, although it is popularly assumed. To the best of my knowledge, the filter does not filter out very much of the material that gets deep into the airways, and the mainstream smoke inhaled by a smoker is far more concentrated than secondhand smoke. (By CO2, did they mean CO?) They cite reference 6 and 7; 6 is a meta-analysis and 7 is a study about cluster headaches and neither supports the statement. I would remove this highly questionable sentence. This in no way minimizes the importance of secondhand smoke as a major source of adverse health effects.

Answer: The sentence and the references were deleted of the manuscript.

Minor comments

1) While the nomenclature of SC, EC, SS, and ES makes sense, it’s REALLY hard to follow because S stands for both Sedentary and Smoker. Yes, it matters if it’s the first or second letter of the pair, but it just makes it much less intuitive to follow along for the reader. Can the authors come up with a similarly informative but more reader-friendly way of denoting the permutations?

Answer: The nomenclature was changed for the groups in the manuscript. Control (C) - without interventions; Exercised (E) - performed RT; Smoker (S) exposed to secondhand cigarette smoke; Exercised Smoker (ES) - exposed to secondhand smoke exposure and performed RT.

2) Abstract line 37: it will be much easier to follow if a comma is added after EC. Otherwise, it’s not immediately clear whether EC belongs to the first comparison or the second. Even then, it’s a confusing line to read.

Answer: The sentence was changed. 

3) The sentence starting on abstract line 41 would be more useful if there was an indication of what the finding means (i.e., tell the reader if reduction of the muscle fiber diameter good or bad).

Answer: The sentence was changed.

4) I rarely comment on keywords, but it seems to me that including “tobacco” as a keyword would be useful.

Answer: The word was included as keyword.

5) Line 57: It’s not accurate to say that the “consumption” of a cigarette is divided into particulate and gas phases; consumption is the verb. Better to say cigarette smoke consists of particulate and gas phases. Actually, it may be a language issue, but the entire paragraph is problematic, and references 6 and 7 seem like odd choices to use here. Are the authors attempting to describe the mainstream smoke that is inhaled through the filter and the sidestream smoke that is generated (not “exhaled”) by the burning tip? This is all irrelevant to gas or particle phases; the phrase on line 58 that the gas phase consists of two types of smoke does not make sense. I suggest that they replace this paragraph (and get rid of those two references) with a simple statement that the smoke inhaled by smokers is mainstream smoke, the smoke generated by the burning tip is sidestream smoke, the secondhand smoke in the air around smokers consists of a mixture of mostly sidestream smoke with some exhaled mainstream smoke, and that secondhand smoke contains high concentrations of noxious components.

Answer: The sentence was substitute by “Secondhand smoke is the combination of two smokes: the mainstream exhaled by smokers plus the burning end of a cigarette (sidestream). It smoke contains more noxious substances, with hundreds toxic, and some can trigger câncer”

6) Line 71: In “…it is fundamental to find tools…”, fundamental should probably be replaced by something like crucial or important.

Answer: The word “fundamental” was substituted for “crucial”, as suggested.

7) Line 74: “pulmonary pressure”: do they mean pulmonary arterial pressure or increased pressure in the lungs?

Answer: The correct is pulmonary arterial pressure.

8) Line 80: I suggest clarifying the first time they say “resistance training” to make it more obvious to readers who aren’t familiar with the topic, especially given that “insulin resistance” shows up a few words later. Perhaps “Recently, exercise in the form of resistance training (RT) was classified…”

Answer: The sentence was changed, as suggested.

9) In Table 1 in the Methods; please add units (presumably g).

Answer: The Table 1 was deleted of the manuscript.

10) Line 237, I think “tricomium” is supposed to be “trichrome”.

Answer: The word was changed.

11) Line 257: what is a “Lee index”? Please clarify in the text. It’s in the figure legend but the reader has not yet encountered that when the term is first used.

Answer: The term was explained in details in Methods section (section 2.5).

12) Line 325: Cigarette smoking is not a drug; cigarettes are the drug.

Answer: The sentence was changed.

13) Line 414: “strength” is misspelled.

Answer: The word was corrected.

14) In Figure 1, the word “cigarettes” is misspelled twice, and it is missing a space in “1x day(4 days)”

Answer: The figure was corrected.

15) The legends for Figures 2-6 include magnification denotations of the form “100x” etc., which are meaningless unless the final print size of the figure is known and not useful when viewing on a monitor. Scale bars are better, and the figures actually have scale bars already but they are much too small to be seen, so it will be a simple matter to make standard visible scale bars based on them. If a magnification number is desired, please state as “10x objective” etc.

Answer: The legends of the figures were changed.

16) In figures 2-5, where quartets of histology panels are shown, it will be much simpler for the reader if panels are labeled as ES, SS, etc. rather than a, b, etc.; it saves the reader from having to look at the legend to understand the figure.

Answer: The labeled was changed for the abbreviation of the groups.

17) I suggest making the asterisks in the bar graphs larger; they are tiny and look like lint on the computer monitor.

Answer: The asterisks are larger.

Reviewer #2: The manuscript by Seraphim et al. aims at determining the effect of resistance training on various parameters in rats exposed to second hand cigarette smoke. Their main conclusion is that the training prevents right ventricle hypertrophy. They provided data that are somehow related to development of right ventricle hypertrophy such as pulmonary artery thickness and lungs histopathology, as well as some supportive information such as body mass gain, food consumption, and masses of the different tissues that they included in their study. Not being an expert on the field, this reviewer cannot say how novel and interesting the findings are for the scientific community. However, there are definitely a number of issues that have to be addressed before the manuscript can be considered for publication.

General observations:

1) This reviewer was invited as an animal experiment ethics specialist. In this respect, the study is fine; there seems to be no ethical concern.

2) Although a language correction has been performed on the manuscript, clarity of the text is still below publishable quality. To illustrate it, this reviewer performed editing of a 1.5 pages section of Materials and Methods. a/ line 120-122: “The smoking rats (SS and ES) were exposed to secondhand cigarette smoke (19) for 16 weeks, which was predetermined to characterize long-term exposure (20).” The second part of the sentence may mean that 16 weeks of exposure is considered as a long-term one. 

Answer: Sixty weeks of exposure means long-term exposure, considering rats can llive during 24 months. 

According the literature (https://doi.org/10.1590/S0102-67202012000100011) each rat month in adulthood is similar to around 2.5 human years. 

So we have:

1 month in adulthood rat corresponding to 2.5 human years and

4 months (16 weeks) in adulthood rat corresponding to around 10 human years,

So, 16 week smoke exposure should be considered long-term exposure.

Anyway, the sentence was removed from the text, stopping at 16 weeks in the paragraph, as following:

“The smoking rats (S and ES) were exposed to secondhand cigarette smoke for 16 weeks”.

b/ line 125: Change “characterizing” to “representing”. 

Answer: The sentence was rewritten.

c/ line 138: Cigarettes never contain carbon monoxide, which is a gas. The indicated 10 mg should be the average dose a smoker receive during a smoke. 

Answer: The sentence was changed for: “Commercial cigarettes were used in the study, containing 10mg of tar, 0.8 mg of nicotine, and 10 mg of carbon monoxide during the combustion”.

d/ line 153-154: “This test was performed only in this moment.” ???? Better delete.

Answer: The sentenced was deleted.

Specific comments:

1) The overall design of the study indicates that more results have been anticipated than have actually been resulted. That creates some problems. The organ masses were properly presented as not significantly altered in line 260. However, according to PLOS ONE’s publishing policies “data not shown” is not permissible. Either the data should be presented, or the observation should be omitted. In this case, I suggest presenting the data as a supplementary table.

Answer: The Table with the organ masses is presented as supplementary material.

2) line 271: On what basis was pulmonary emphysema diagnosed? Put it in the Methods section.

Answer: The sentence was added in Methods section: “Pulmonary emphysema was diagnosed when the alveolar spaces were enlarged and the alveolar septa retracted”.

3) Fig 4: Exercise increased right ventricle thickness in non-smoker rats while decreased it to BELOW THE CONTROL LEVEL in smoker ones. These data are very hard to understand.

Answer: In fact, smoking caused increase of the right ventricle thickness (see S group). Resistive training did not change the thickness of left and right ventricles nor in non-smoking (E) nor in smoking (ES) rats.

4. The number of Kupffer cells in the liver is presented in Fig 6. Either move this figure to the supplementary section or explain in the Discussion section the relevance of the data presented in it to right ventricle hypertrophy.

Answer: The paragraph was changed correlating liver and heart diseases and the figure 6 was maintained in the manuscript.

5. The Discussion section is poor. Please, interpret the data presented in the Results section regarding right ventricle hypertrophy and related studies on the field. Also, please avoid drawing conclusions on unsupported data.

 Answer: Thanks for the suggestion, changes were included in the Discussion Section.

---

## [Decision Letter · Decision Letter 1]

29 Apr 2020

PONE-D-19-31709R1

Resistance training prevents right ventricle hypertrophy in rats exposed to secondhand cigarette smoke

PLOS ONE

Dear Dr Seraphim,

Thank you for submitting your manuscript to PLOS ONE. After careful consideration, we feel that it has merit but does not fully meet PLOS ONE’s publication criteria as it currently stands. Therefore, we invite you to submit a revised version of the manuscript that addresses the points raised during the review process.

While both reviewers have said that the manuscript is improved, there are still a lot of problems, including sloppy presentation of the results and various inconsistencies in the manuscript.

I am giving you one more chance to clean up this manuscript.  I will then send it back to the reviewers.  If they are happy, it will be accepted.  Otherwise, it will be rejected.

I strongly suggest that you have some people who are not authors on this paper carefully review everything before you resubmit the paper.  It is not the reviewers' job to write the paper for you.

We would appreciate receiving your revised manuscript by Jun 13 2020 11:59PM. To enhance the reproducibility of your results, we recommend that if applicable you deposit your laboratory protocols in protocols.io, where a protocol can be assigned its own identifier (DOI) such that it can be cited independently in the future. For instructions see: http://journals.plos.org/plosone/s/submission-guidelines#loc-laboratory-protocols

We look forward to receiving your revised manuscript.

Kind regards,

Stanton A. Glantz

Academic Editor

PLOS ONE

Reviewers' comments:

Reviewer's Responses to Questions

**Comments to the Author**

1. If the authors have adequately addressed your comments raised in a previous round of review and you feel that this manuscript is now acceptable for publication, you may indicate that here to bypass the “Comments to the Author” section, enter your conflict of interest statement in the “Confidential to Editor” section, and submit your "Accept" recommendation.

Reviewer #1: (No Response)

Reviewer #2: (No Response)

2. Is the manuscript technically sound, and do the data support the conclusions?

Reviewer #1: Partly

Reviewer #2: Partly

3. Has the statistical analysis been performed appropriately and rigorously? 

Reviewer #1: Yes

Reviewer #2: I Don't Know

4. Have the authors made all data underlying the findings in their manuscript fully available?

Reviewer #1: Yes

Reviewer #2: Yes

5. Is the manuscript presented in an intelligible fashion and written in standard English?

Reviewer #1: Yes

Reviewer #2: No

6. Review Comments to the Author

Reviewer #1: First of all, an important note: there are two versions of the revised manuscript, one clean and the other with changes tracked, but the two version are not identical. I realized in the middle of reviewing that at least one change made in the tracked changes version is not changed in the untracked version (see SS in line 304 of the clean version that has been changed to S (with the line through the deleted extra S) in corresponding line 314 of the tracked changes version. I don’t know if more slipped through, but it means that the tracked changes version is the one that should be used, or else some changes might become unchanged.

The authors have made various changes to respond to previous critiques. I think the new nomenclature for the four groups is much clearer than the previous one; a good solution. Note that the changes in group designations all show up in the changes tracked version but as mentioned above, at least one of them is not in the untracked version.

One remaining issue that we still need to fix is the lack of clarity about the smoke exposure system, despite the authors having added more details. The problem is that with a custom-built system, details of how the system works are important so that the results can be put in context of other smoke exposure literature. From the new description added, it’s still not clear to me how the smoke is being generated, and they refer to one of their previous papers but it is in Portuguese. Another new reference is to a review article about various effects of smoke on the lung; there’s no procedural detail there that is relevant to this paper. Most studies that produce secondhand smoke follow some variation of a pretty consistent workflow: air is sucked through a cigarette by some mechanism and according to a specific puff protocol (e.g. ISSO, Health Canada Intense), and the smoke from the burning tip is collected. Cigarettes are frequently pre-equilibrated to consistent humidity before use. These conditions provide not just consistency between experiments, but also the ability to place the results in context of the rest of the body of smoke research.

For example, there’s no mention in the description of air going through the cigarettes; just that the cigarettes are in a compartment and air goes from that compartment to the rats. Does that mean the cigarettes are sitting on the floor of the compartment simply smoldering in place, with no periodic puffing? If so, it would not invalidate the system, but it also would not be consistent with how such studies are typically accomplished and that should be specifically mentioned. The chemistry is different if the cigarette is smoldering in an ashtray for 10 minutes vs. if someone is actually smoking it and puffing occasionally. Please show a photograph or a diagram, and make the air path clear. It’s also not clear to me what they mean when they say that the dose of smoke exposure is equivalent to 10 to 20 cigarettes for a chronic human smoker, because their model is secondhand exposure; that can’t be directly compared with the number of cigarettes being smoked by people.

The other changes have all been made satisfactorily. I noted a misspelling in the newly added material, “simoustaneouly” in the second-to-last paragraph of the discussion (line 460 in the tracked changed version).

Reviewer #2: Although the English of the manuscript has improved (at least the first 15 pages), it is still far from tolerable (at least by this reviewer). The manuscript either has not been revised grammatically, or the proof-reader has made a very poor job. This reviewer could not evaluate the revised manuscript because of the quality of English. Proof-reading is outside the reviewers' responsibilities, although the other reviewer very kindly has also suggested a number of such mistakes in the first round. In an original submission, the mistakes could be tolerated, however, not in a revised manuscript. This reviewer has got as far as the top of page 16 before quitting, and does not want to waste his/her time on submissions that do not reflect due care from the authors' side.

page 14, line81-83: Clarify the sentence. Most probably the first comma is unnecessary. Regardless, as presented, the third part of the sentence does not confer comprehensible meaning.

page 14, line 98: Change “monitoring” to “monitored”. Change “intake food” to “intake of food”.

page 15, line 120: Change “04” to “4”, then do the same throughout the text.

page 15, line 121: Delete “only”.

7. PLOS authors have the option to publish the peer review history of their article (what does this mean?). If published, this will include your full peer review and any attached files.

Reviewer #1: No

Reviewer #2: No

---

## [Author Response · Author response to Decision Letter 1]

12 Jun 2020

We would like to start by acknowledging the thoroughness of the revision and the importance of the reviewer’s comments that greatly improved our manuscript.

The answers for the reviewers’ questions are below, as following:

Reviewer #1: 

Comment 1) First of all, an important note: there are two versions of the revised manuscript, one clean and the other with changes tracked, but the two version are not identical. I realized in the middle of reviewing that at least one change made in the tracked changes version is not changed in the untracked version (see SS in line 304 of the clean version that has been changed to S (with the line through the deleted extra S) in corresponding line 314 of the tracked changes version. I don’t know if more slipped through, but it means that the tracked changes version is the one that should be used, or else some changes might become unchanged.

Response to comment 1 -: I would like to apologize because the final file of the Manuscript with untracked changes was really different of the version with tracked changes. I have made a mistake during the submission process, changing the files. I apologize about this mistake. But I am sure that this error did not compromise the relevance of the manuscript and the comprehension of the text in the manuscript.

Comment 2) The authors have made various changes to respond to previous critiques. I think the new nomenclature for the four groups is much clearer than the previous one; a good solution. Note that the changes in group designations all show up in the changes tracked version but as mentioned above, at least one of them is not in the untracked version.

Answer to comment 2 – We agreed with the reviewer. The designation of the groups was much better. This time, we took care about using the same designation for all groups in the untracked version. We have checked all nomenclature before sending the new version.

Comment 3) A) One remaining issue that we still need to fix is the lack of clarity about the smoke exposure system, despite the authors having added more details. The problem is that with a custom-built system, details of how the system works are important so that the results can be put in context of other smoke exposure literature. From the new description added, it’s still not clear to me how the smoke is being generated, and they refer to one of their previous papers but it is in Portuguese. Another new reference is to a review article about various effects of smoke on the lung; there’s no procedural detail there that is relevant to this paper. Most studies that produce secondhand smoke follow some variation of a pretty consistent workflow: air is sucked through a cigarette by some mechanism and according to a specific puff protocol (e.g. ISSO, Health Canada Intense), and the smoke from the burning tip is collected. Cigarettes are frequently pre-equilibrated to consistent humidity before use. These conditions provide not just consistency between experiments, but also the ability to place the results in context of the rest of the body of smoke research.

3 B) For example, there’s no mention in the description of air going through the cigarettes; just that the cigarettes are in a compartment and air goes from that compartment to the rats. Does that mean the cigarettes are sitting on the floor of the compartment simply smoldering in place, with no periodic puffing? If so, it would not invalidate the system, but it also would not be consistent with how such studies are typically accomplished and that should be specifically mentioned. The chemistry is different if the cigarette is smoldering in an ashtray for 10 minutes vs. if someone is actually smoking it and puffing occasionally. Please show a photograph or a diagram, and make the air path clear. 

3 C) It’s also not clear to me what they mean when they say that the dose of smoke exposure is equivalent to 10 to 20 cigarettes for a chronic human smoker, because their model is secondhand exposure; that can’t be directly compared with the number of cigarettes being smoked by people.

Answer to comment 3 - A) The reference 20 in Portuguese was changed for another reference published in English by the same group: Garcia BC, Bonfim MR, Camargo RCT, Souza DRS, Abreu LC, Filho, JCSC. Effects of passive smoking associated with physical exercise in the skeletal muscle of rats during pregnancy and lactation. Int J Morphol. 2015;33:497-506. This study is from a research group that used the same chamber of exposure. We also added the reference # 18 in the line 138 (tracked version) or line 130 (untracked version), to justify the use of the model of chamber of exposure to cigarette smoke. But, in both articles, there is no adittional detail how the air flow was generated and the direction of this flow. As it is a custom-built system, we agreed with the reviewer, and more details are crucial to understand how the system works. In this new version we are adding the Fig. 1 showing the detail of the air flow inside the chamber and in the text we are describing the system more detaily. Because of this, we had to change the number of all figures in the manuscript as well.

Answer to comment 3 - B) During the exposure, the cigarettes were lighted and were left burning with no periodic puffing. Cigarettes were not pre-equilibrated to consistent humidity before the use. But 4 animals per time were let in the exposure chamber. And in the next day, we changed the sequence of the cages that would begin the exposure. A more detailed description was added in the Methods (lines 131-144 in the untracked version).

Answer to comment 3 - C) We agreed with the reviewer when he said that the dose of smoke exposure can’t be directly compared with the number of cigarettes being smoked by people, because of the model is secondhand exposure. The sentence was withdrawn (see lines 131-133 in the tracked manuscript).

Comment 4) The other changes have all been made satisfactorily. I noted a misspelling in the newly added material, “simoustaneouly” in the second-to-last paragraph of the discussion (line 460 in the tracked changed version).

Answer to comment 4 – The word was corrected to simultaneously (line 432).

Reviewer #2: Comment 1) Although the English of the manuscript has improved (at least the first 15 pages), it is still far from tolerable (at least by this reviewer). The manuscript either has not been revised grammatically, or the proof-reader has made a very poor job. This reviewer could not evaluate the revised manuscript because of the quality of English. Proof-reading is outside the reviewers' responsibilities, although the other reviewer very kindly has also suggested a number of such mistakes in the first round. In an original submission, the mistakes could be tolerated, however, not in a revised manuscript. This reviewer has got as far as the top of page 16 before quitting, and does not want to waste his/her time on submissions that do not reflect due care from the authors' side.

Answer: -: We would like to apologize about the grammar mistakes. We have sent the certificate of the English review made in the first version of the manuscript in attachment. After this, we added some new sentences in the manuscript to answer the reviewers, and to adequate the manuscript. However, we did not submit under an English review again, but we believe in that our inclusion in the text did not compromise the relevance of the study. We really believe in our results in this manuscript. We really believe in that the manuscript is adequate now.

Comment 2) - page 14, line81-83: Clarify the sentence. Most probably the first comma is unnecessary. Regardless, as presented, the third part of the sentence does not confer comprehensible meaning. We really believe in the manuscript is adequate now.

Answer: The third part of the setence explains the difference between anaerobic and aerobic training. If the pauses of recovery among the series were not done, the exercise would become aerobic. High-intensity and short-term duration with pauses among series characterize anaerobic exercise. The sentence was modified for better comprehension (see lines 82-84 in the both untracked and tracked versions).

Comment 3) -page 14, line 98: Change “monitoring” to “monitored”. Change “intake food” to “intake of food”. 

Answer: all changes were done.

Comment 4) -page 15, line 120: Change “04” to “4”, then do the same throughout the text. 

Answer: all changes were done

Comment 5) -page 15, line 121: Delete “only”.

 Answer: all changes were done.

---

## [Decision Letter · Decision Letter 2]

20 Jul 2020

Resistance training prevents right ventricle hypertrophy in rats exposed to secondhand cigarette smoke

PONE-D-19-31709R2

Dear Dr. Seraphim,

We’re pleased to inform you that your manuscript has been judged scientifically suitable for publication and will be formally accepted for publication once it meets all outstanding technical requirements.

Kind regards,

Stanton A. Glantz

Academic Editor

PLOS ONE

Additional Editor Comments (optional):

Reviewers' comments:

Reviewer's Responses to Questions

**Comments to the Author**

1. If the authors have adequately addressed your comments raised in a previous round of review and you feel that this manuscript is now acceptable for publication, you may indicate that here to bypass the “Comments to the Author” section, enter your conflict of interest statement in the “Confidential to Editor” section, and submit your "Accept" recommendation.

Reviewer #1: All comments have been addressed

Reviewer #2: All comments have been addressed

2. Is the manuscript technically sound, and do the data support the conclusions?

Reviewer #1: Yes

Reviewer #2: Yes

3. Has the statistical analysis been performed appropriately and rigorously? 

Reviewer #1: Yes

Reviewer #2: Yes

4. Have the authors made all data underlying the findings in their manuscript fully available?

Reviewer #1: Yes

Reviewer #2: Yes

5. Is the manuscript presented in an intelligible fashion and written in standard English?

Reviewer #1: Yes

Reviewer #2: Yes

6. Review Comments to the Author

Reviewer #1: (No Response)

Reviewer #2: All my comments have been adequately addressed, including those ones of the previous revision round.

7. PLOS authors have the option to publish the peer review history of their article (what does this mean?). If published, this will include your full peer review and any attached files.

Reviewer #1: No

Reviewer #2: No